# "My father insisted that I have the baby but not in his house": Adolescent pregnancy, social exclusion and (dis)empowerment of girls in an urban informal settlement in Kenya

**Beryl Nyatuga Machoka** *, **Caroline W. Kabiru**, **Anthony Idowu Ajayi**

Sexual, Reproductive, Maternal, Newborn, Child and Adolescent Health (SRMNCAH) Unit, African Population and Health Research Center, Nairobi, Kenya

* berylninanyatuga@gmail.com

**Data Availability Statement:** All data used in the analysis are contained in the manuscript. Full transcripts of the interviews could be accessed via:

## Abstract

While the drivers of adolescent pregnancy are widely studied, few studies have examined the social exclusions associated with early and unintended pregnancy. Drawing data from a larger mixed methods study on the lived experiences of pregnant and parenting adolescents and guided by Amatya Sen's social exclusion framework, this qualitative explanatory study examines how poverty and the contestation around girls' access to comprehensive sexuality education hinder them from preventing unintended pregnancy. It also examines why adolescent pregnancy further results in girls' social exclusion with implications for their health and socioeconomic (dis)empowerment. We drew on data from in-depth interviews with purposively selected pregnant and parenting adolescents aged 15 to 19 (n = 22) and parents (n = 10), and key informant interviews with teachers (n = 4), policymakers (n = 3), community leaders (n = 6) non-governmental organization representatives (n = 2), and health workers (n = 4). Through inductive and deductive thematic analysis, we found that poverty and lack of access to contraceptive information and services contributed to girls' vulnerability to early unintended pregnancies. Becoming pregnant exacerbated girls' social exclusion, which is characterized by self-isolation, being disowned by their families, or forced to drop out of school. Shame, stigma, and discrimination of girls made girls seek antenatal care late. They also failed to complete the recommended number of antenatal care visits. Marginalization of pregnant and parenting girls results in them being out of school, vocational training, and employment, as well as experiencing mental distress. The analysis shows the social exclusion of girls is cyclical, beginning before their pregnancy and continuing into pregnancy and post-pregnancy. Their social exclusion has negative implications for their health and socioeconomic empowerment. Interventions to address adolescent childbearing should holistically address the social exclusion that predisposes girls to unintended pregnancy and that follows during and post-pregnancy, as tackling this exclusion is key to improving their health and socioeconomic well-being.

http://microdataportal.aphrc.org/index.php/catalog/149/data-dictionary/F8?file_name=Kenya%20PPA%20Dataset".

**Funding:** This study was funded by a grant from the African Regional Office of the Swedish International Development Cooperation Agency, Sida Contribution No. 12103 for the African Population and Health Research Center's Challenging the Politics of Social Exclusion project. The funders had no role in study design, data collection and analysis, decision to publish, or preparation of the manuscript.

**Competing interests:** The authors have declared that no competing interests exist.

## Introduction

Adolescent pregnancy—often unintended and outside wedlock—is widely recognized as a public health problem and social development challenge. While many scholars, development partners, and non-government organizations argue that equipping young people with comprehensive sexuality education (CSE) is key to reducing adolescent pregnancy [1], anti-rights movements oppose CSE, countering that it sexualizes children and promotes indiscriminate sex [2, 3]. Both voices permeate the African continent where one in four girls begin childbearing before the age of 18 [4]. At the center of this contestation are millions of young people who continue to become pregnant yearly on the continent [5]. For many of these girls, early pregnancy means the end of their educational goals [6, 7] and in extreme cases, their death [8–10]. Research indicates that adolescents face a higher risk of childbirth and pregnancy complications [11, 12], and pregnancy-related complications are the leading cause of death among adolescent girls in sub-Saharan Africa [13]. Adolescent girls are also the most likely to experience severe complications of unsafe abortion, including death. Babies born to adolescent mothers also face an increased risk of neonatal mortality, low birth weight, and preterm birth [11].

Girls who experience early pregnancy face stigma, and rejection by peers, family, and community [14–18]. The stigma is driven by social norms around sexual purity for girls [19] and results in self-isolation, school dropout, avoidance of antenatal care [12, 20], poor mental health [17, 21] and disempowerment [22–24]. In Kenya where 15% of adolescent girls start childbearing before the age of 19 [25], research reveals that social stigma against adolescent pregnancy—including avoidance, ridicule, and mistreatment—forces girls to hide or relocate to new residences [16, 18]. Pregnancy-related stigma results in poor health outcomes, low educational attainment, and economic inactivity that expose girls to dependence and loss of individual agency in society [16, 18, 26]. Of particular concern are girls in urban slum settlements, which are characterized by dense populations, inadequate infrastructure, and limited access to essential services [16]. Compared to their counterparts in non-slum areas, girls in urban Kenyan slums face heightened vulnerabilities to early pregnancy due to high levels of sexual violence, insecurity, socio-economic disparities, limited access to education, and contraceptive and reproductive health information [16, 27, 28].

The literature is replete with studies on drivers of adolescent pregnancy [29–33] and a few studies have explored stigma associated with early childbearing [34, 35]. However, there is little focus on how the social exclusion of girls results in adolescent pregnancy and how their continued exclusion during and after pregnancy affects girls' health and socioeconomic wellbeing. We define socioeconomic well-being as pregnant and parenting girls' overall economic and social status including, income, education, employment, access to basic services or materials. This study seeks to address three questions: 1) How does social exclusion contribute to adolescent pregnancy in low-income informal settlements? 2) How does early pregnancy amplify the social exclusion experienced by pregnant and parenting girls? 3) What are the implications of social exclusion for the education and socioeconomic empowerment of pregnant and parenting girls? This study draws on qualitative interviews with pregnant and parenting adolescents, their parents, teachers, health workers, and community leaders to examine why poverty and the contestation of girls' access to contraceptives hinder them from preventing unintended pregnancy. The study also examines why adolescent pregnancy amplifies girls' social exclusion and its implications for their health and economic (dis)empowerment.

### Amartya Sen's social exclusion framework and adolescent pregnancy

Amartya Sen's social exclusion framework [36] guided this study and helps to understand the nature and intersectional dimensions of social exclusion in the context of adolescent

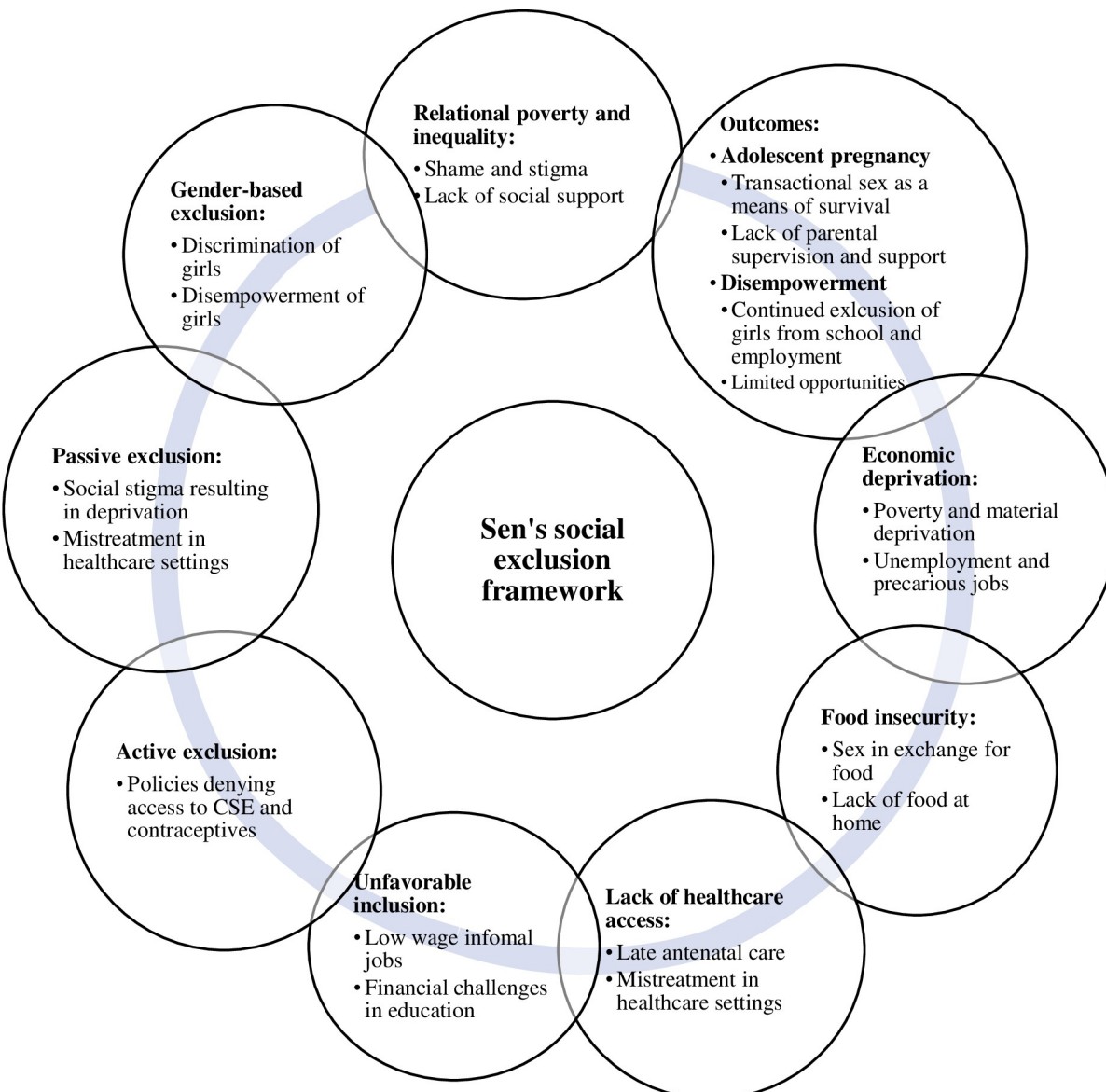

**Fig 1. Adolescent pregnancy and disempowerment of girls within Amartya Sen's social exclusion framework. Center:** Represents Amartya Sen's Social Exclusion Framework, the lens through which adolescent pregnancy and social exclusion is analyzed. **Surrounding circles:** Depict the various ways in which adolescent pregnancy aligns with and impacts the dimensions of social exclusion as outlined in Sen's framework. **Connecting line:** Demonstrates the cyclical and interconnected relationships among the various elements of adolescent pregnancy and social exclusion.

pregnancy (Fig 1). Amartya Sen's social exclusion framework emphasizes the importance of relational problems and the progressive deprivations that prevent full participation in society. Since Sen theorizes social exclusion as a cycle, we incorporate an intersectional perspective [37] to enrich our understanding of how various forms of disadvantage cause deprivation. Intersectionality is an analytical framework in sociology that explores the intersections of different social and political identities, leading to distinct experiences of discrimination and privilege. It emphasizes that various aspects of identity are interconnected intricately, leading to advantages or disadvantages, and helping to understand the complexity of human experience. Social exclusion is a multidimensional process characterized by progressive deprivations that

prevent individuals from participating fully in their societies [36]. This framework has been employed in previous youth research in India [38] and Ethiopia [39].

Amartya Sen's framework emphasizes that both deprivation and social exclusion are major factors that perpetuate a series of disadvantages. According to Sen, social exclusion is a cycle characterized by factors such as relational poverty, inequality, food insecurity, gender-based exclusions, lack of access to healthcare, and unfavorable inclusion in the labor market. These factors collectively limit people from fully participating in society and deprive them of important capabilities and lead to a poverty trap. For example, relational poverty and inequality happen when individuals feel ashamed to appear in public due to exclusionary practices or social stigma. Similarly, food insecurity occurs when people lack the means to purchase or grow sufficient food, thus undermining their health and well-being.

With this perspective, Sen also highlights the significance of tackling unfavorable inclusion —where people are formally included in political, economic, or social systems but do not receive the full benefits inclusion ideally provides. Examples of unfavorable inclusion include individuals working in low-wage or informal jobs without job benefits or security, students from disadvantaged backgrounds facing financial challenges that limit educational attainment, or inadequate healthcare access due to lack of information or money.

Research conducted among girls on the continent has linked poverty and material deprivation to transactional sex and early childbearing [40]. Studies have shown that adolescent pregnancy disproportionately affects girls from poor families and those living in low-income neighborhoods characterized by high unemployment, violence, and precarious work [41, 42]. Girls whose parents are poor, unemployed, or work in precarious jobs lack basic needs such as food, shelter, clothing, money, sanitary pads, and personal effects, and are significantly vulnerable to early childbearing. Men prey on these vulnerable adolescent girls, promising to meet these basic needs in exchange for sex.

With shrinking opportunities, more and more parents are excluded from decent work with sufficient income. We argue that the social exclusion of parents of adolescent girls from decent jobs that earn living wages enough to escape the poverty trap leads girls to engage in survival sex resulting in early and unintended pregnancy. The exclusion of willing and able parents from decent living and meaningfully paying jobs is the largest pull to poverty [43, 44]. Parents' lack of jobs or precarious jobs significantly limit their ability to provide for their children. Likewise, precarious work means parents work long hours in low-paying jobs and significantly spend time away from their children. This limits their ability to supervise their children and deter them from engaging in survival sex. Studies have shown that lack of parental support and supervision are associated with transactional sex and adolescent pregnancy [16]. Adolescent girls born to middle-class to rich parents become pregnant later and are largely shielded from the negative consequences of early childbearing [45].

Amartya Sen distinguishes between passive and active exclusion. He theorizes that active exclusion results from deliberate actions, such as policies intentionally denying young people access to CSE or contraceptives. Research has shown that young people value privacy when seeking sexual and reproductive healthcare services [46]. In low-income areas, adolescents often face barriers such as inability to afford healthcare services and inadequate healthcare infrastructure [47]. Therefore, enacting policies that restrict their access to contraceptives can further increase their vulnerability to unintended pregnancies. In Kenya, adolescents' access to contraceptives is expressly restricted through the Child Rights Act and Reproductive Health policy given the requirement for parental consent before services are provided. Requiring parental consent before sexual and reproductive health services are provided to adolescents who seek these services could ultimately mean that adolescents forfeit accessing services altogether. The implication of excluding adolescents from critical preventive sexual and

reproductive health information and services is that they are disempowered in preventing early and unintended pregnancy. Punitive policy that expressly prohibits pregnant and parenting girls from formal schooling, such as in Tanzania and Sierra Leone—although now reversed [48]—is another example of active exclusion.

Passive exclusion, on the other hand, lacks a calculated intent to exclude since deprivation is the outcome of social processes. For example, health workers might actively encourage girls to seek care in health facilities but mistreat them during their visits, causing passive exclusion. Differentiating between active and passive exclusion enhances understanding of how policies and practices at the community, national, and international levels contribute to the deprivation of capabilities.

Another concept highlighted by Sen is that exclusion is inherently depriving and can cause more deprivations. For example, being unable to socialize can initiate a chain of deprivations including limited economic opportunities stemming from social relations. This concept facilitates the analysis of the causal chain of social exclusion in contexts where exclusion is characterized by multifaceted factors. Despite the enactment of school reentry policies in many African countries, most adolescent mothers are out of school [49]. This is evidence of how adolescent pregnancy results in the further exclusion of girls. The continued social exclusion of girls jeopardizes their education dreams and perpetuates gender inequality. Girls who have early and unintended pregnancies encounter numerous financial and social barriers that force them to discontinue their formal education [50]. Low educational achievement among girls constrains their economic potential and ability to make meaningful contributions to society, ultimately hindering the progress toward achieving gender equality [51]. What is more, the circle continues even for the children of adolescent mothers, leading to an inter-generational poverty trap [52]. Breaking this chain of poverty requires addressing the social exclusion leading to early pregnancy and post-pregnancy exclusion.

Interventions that support adolescent mothers are critical to addressing post-pregnancy exclusion. However, critics argue that supporting adolescent mothers could unintentionally encourage more girls to become pregnant. Such is the case in South Africa, where the child support grant system, which was initiated in 1998 to aid primary caregivers of children under 18 years, and makes monthly payments to 12.8 million beneficiaries [53], has been criticized for fueling adolescent childbearing due to its cash benefits [54]. This criticism offers evidence that society views these adolescent mothers as "undeserving" of support. This perspective aligns with research findings that when society characterizes an individual as "undeserving" it tends to attribute their circumstances to personal choices, often overlooking the potential effect of broader societal factors in shaping their situation [55].

By adopting an intersectional perspective, we examine interconnected factors contributing to the exclusion of pregnant adolescents. This approach aims to inform policies and interventions aiming to address the complex needs of adolescent girls and promote their social inclusion. This framework will be employed to analyze how social exclusion contributes to adolescent pregnancy and perpetuates cycles of poverty and disempowerment among pregnant and parenting adolescent girls.

## Methods

### Study design

We adopted a qualitative explanatory design [56] to gain insight into participants' experiences of social exclusion leading to their early pregnancy and resulting from their becoming pregnant through the use of qualitative interviews and thematic analysis. Given the analytical goal of comprehending the effects of social exclusion on adolescent pregnancy, as well as why early

childbearing might intensify social exclusion, a qualitative explanatory design was appropriate. Qualitative data analyzed in this study were drawn from a larger cross-sectional mixed methods study focusing on understanding the lived experiences of pregnant and parenting adolescents in an urban informal settlement in Nairobi, Kenya. However, only the qualitative data are relevant to this study's objectives.

## Study setting

Data were collected in Korogocho, a low-income urban informal settlement, between November 3[rd] 2022 and December 16[th] 2022. The selection of Korogocho was intentional, partly influenced by its high prevalence of adolescent pregnancy and because it has been a site under the Nairobi Urban Health and Demographic Surveillance System (NUHDSS) for over two decades. Korogocho's demographic and socio-economic characteristics provide context for examining the intersection of poverty, social exclusion, and early pregnancy among adolescents. In Nairobi, nearly a third of the urban slum population is comprised of young people [57]. According to the data from the NUHDSS, Korogocho is highly congested and the fourth largest informal settlement in the city [58]. Slum residents experience the worst form of socio-economic and health deprivation in Kenya [58]. In Kenya, 15% of girls become pregnant before the age of 19 [25]. However, adolescents living in impoverished neighborhoods are more likely to have early and unintended pregnancies [28, 59]. Approximately 41% of adolescent girls in Nairobi's informal settlements have experienced early pregnancy [60]. Adolescent pregnancies in Kenya are partially driven by high poverty levels as about 8% of girls in the uppermost wealth quantile experience early childbearing compared to 21% in the lowest wealth quantile [25].

## Participants and recruitment

We collaborated with a community-based youth empowerment organization (CBO) focused on addressing the vulnerabilities of young people in the study area to recruit the participants. None of the authors worked in the CBO. Through this partnership, outreach staff from the CBO identified and purposively sampled [61] adolescent girls, parents and key informants who met the inclusion criteria. During sampling, they utilized physical tracing in the community, referrals, and existing records to identify and invite the first and second groups of participants to take part in the study.

The inclusion criteria for the first group of participants required them to be adolescent girls aged 10 to 19 who were either pregnant or actively parenting and lived in the community during the study period. Consent for participants aged 18 and above was sought directly from them, while for those under 18 years, consent was obtained from their parents. As qualitative data collection was part of a larger mixed-methods survey, the purposively sampled girls did not take part in the quantitative study because both qualitative and quantitative data collection were conducted concurrently. As shown in Table 1, the interviewed girls' ages ranged from 15 to 19 years. Two of them had experienced repeat pregnancy and most of them were out of school at the time of the interview. One was cohabiting while others described their relationship status as single. Most of the girls (15 out of 22) were unemployed; five worked as domestic help and two were self-employed. Participants were diverse, ensuring we captured the narratives of both pregnant adolescents and those who had transitioned into parenting roles. Two of them were currently pregnant and had not given birth while twenty of them had already given birth.

The second and third groups of participants were included to enhance the understanding of adolescent pregnancy, social exclusion, and the disempowerment of girls. The second group

**Table 1. Socio-demographic characteristics of pregnant and parenting adolescent girls.**

| Girls' socio-demographics | Number of participants |
|---|---|
| Age | 15 years (n = 1) |
| | 16 years (n = 2) |
| | 17 years (n = 6) |
| | 18 years (n = 5) |
| | 19 years (n = 8) |
| Pregnancy intendedness | Pregnancy was unintended (n = 22) |
| Pregnancy status | Currently pregnant but has never given birth (n = 2) |
| | Has given birth (n = 20) |
| Number of pregnancies | One (n = 20) |
| | Two (n = 2) |
| Education | Dropped out of primary school (n = 4) |
| | Dropped out of secondary school (n = 13) |
| | Completed primary education (n = 3) |
| | Completed secondary education (n = 1) |
| | Still in secondary school (n = 1) |
| Marital status | Single (n = 21) |
| | Cohabiting (n = 1) |
| Employment | Unemployed (n = 15) |
| | Domestic and casual work: washes clothes and dishes (n = 1), washes clothes and hairdressing (n = 1) washes clothes only (n = 1) washes clothes and does packaging work in a bakery/market (n = 2) |
| | Self-employed: sells shoes (n = 1), makes and sells bags, and owns a hair salon (n = 1) |
| Partners' age | 17 to 18 years (n = 17) |
| | 20 and older (n = 5) |
| Partners' occupation | Unemployed (n = 7) |
| | Casual work: plumbing (n = 1), fetching water (n = 1), construction work (n = 3), washing cars (n = 1), farm work (n = 1), garbage collection (n = 1) |
| | Employed: "Engineer" (n = 1), an employee at a cereal store (n = 1), an employee at a motor vehicle company (n = 1) |
| | Self-employed: selling clothes (n = 1), motorbike operator (boda-boda) (n = 3), mechanic (n = 1), owns video and cybercafé and MPESA shop (n = 1) |
| Family structure | Lives with relatives: aunt (n = 1), uncle (n = 1), grandmother (n = 1), sister and sister's husband (n = 1) |
| | Lives with the boyfriend (n = 2) |
| | Lives with both parents and siblings (n = 3) |
| | Lives with the mother, step-father, and siblings (n = 2) |
| | Lives with the mother and siblings (n = 8) |
| | Lives with the husband (n = 1) |
| | Lives with a female friend (n = 1) |
| | Lives alone (n = 1) |
| Schooling and pregnancy | Still in school when they got pregnant (n = 17) |
| | Had dropped out of school when they got pregnant (n = 5) |

included parents and legal guardians. Their inclusion criteria required individuals to be legal guardians or parents of pregnant or parenting adolescent girls residing in the study site.

The third group consisted of key informants such as teachers, policymakers, community leaders, health workers, and NGO representatives. To identify and recruit them, the CBO

conducted physical searches throughout the community, cross-referenced their existing records, and leveraged contacts within their professional network. The inclusion criteria required that they either lived or worked in the study site and had direct interactions with the community's pregnant and parenting girls and their parents.

## Data collection

The study employed interview guides (S1 Text) with open-ended questions in Kiswahili to explore the lived experiences of pregnant and parenting girls and, in particular, how social exclusion contributed to adolescent pregnancy and disempowerment of girls. We conducted in-depth interviews with girls (n = 22), parents (n = 10), and key informant interviews with teachers (n = 4), policymakers (n = 3), community leaders (n = 6) non-governmental organization representatives (n = 2), and health workers (n = 4). We were more interested in obtaining and triangulating diverse perspectives of participants as opposed to achieving information saturation or redundancy. Nonetheless, the sample included was enough to provide diversity and depth of information to achieve thematic saturation.

We recruited and trained four female and one male qualitative research assistants over a five-day period. The training encompassed the study overview, objectives, interviewer roles, the consent process, research ethics, and role-playing exercises using the interview guides. The research team prepared the interview guides and distributed them to the interviewers. A pilot study was carried out on the fourth day of training, and the insights gained were used to fine-tune the interview guides.

During data collection, the research team approached individuals who met the inclusion criteria in face-to-face meetings, introduced themselves, explained the study's objectives, and sought voluntary participation. Only those willing and mentally capable of responding in Kiswahili were interviewed. Research assistants conducted face-to-face interviews in Kiswahili. Interviews were conducted with participants who consented to be interviewed and recorded. Notes were taken alongside audio recordings. Research assistants were attentive to the participants' distress cues, including visible shaking, crying, long pauses between conversations, and changes in tone. Accordingly, they paused the interviews when necessary to allow participants to regain composure. In one case of extreme distress, the research team referred the participant to a professional counselor and excluded her from the study to prioritize her well-being. The interviews lasted one hour on average and were conducted in private spaces set aside in the community by the research team. Minimal interruptions were encountered during the interviews with girls as they held their babies when interviewed. To ensure data validity, audio recordings were transcribed verbatim by bi-lingual translators. No repeat interviews were conducted.

## Data analysis

The data were analyzed using thematic analysis drawing upon Amartya Sen's social exclusion framework, and grounded in interpretive phenomenological analysis. This approach allows us to evaluate the participants' lived experiences and provide an interpretation of their perspectives on adolescent pregnancy and social exclusion. Within this method, we applied inductive and deductive reasoning as part of our analytical approach [62] to explore themes related to adolescent pregnancy, social exclusion, and (dis) empowerment of girls drawing from Amatya Sen's social exclusion framework [36]. Inductive analysis was conducted initially to allow themes to naturally emerge from the data. Once key themes and patterns were identified, deductive analysis was applied to further examine these themes in the context of Amartya Sen's social exclusion theoretical framework.

Bilingual interviewers transcribed and translated the audio recordings into English, ensuring accuracy and completeness. We thoroughly reviewed all transcripts before transferring them to NVivo for coding and analysis. BNM read the transcripts for familiarity and coded the data. BNM reviewed the codes through an iterative process to ensure the data was captured accurately. Similar codes were grouped under sub-themes and applicable themes derived from the data. BNM then re-read each transcript and organized the transcripts into themes agreed upon by all authors. AIA randomly appraised the themes and sub-themes captured in the coding map (Fig 2) to guarantee the credibility of the analysis. The themes captured major patterns, responses, and concepts relevant to the study objectives. BNM and AIA used the coding map as a guide for analysis and results interpretation.

## Rigor and trustworthiness

Throughout the study, we ensured rigor and trustworthiness to guarantee the integrity and usefulness of the findings. We were meticulous, transparent, and critical, continually scrutinizing our methods and results while seeking alternative explanations. To verify the completeness and accuracy of the data, we compared transcript samples with the audio recordings. We also triangulated the perspectives of various study participants and examined both positive and negative cases. Utilizing NVivo software for data analysis, audio recording, and transcription provided a comprehensive audit trail for the data collection and analysis processes.

## Ethical consideration

We obtained ethics approval from the AMREF Health Africa's Ethics and Scientific Review Committee (ESRC) and the African Population and Health Research Center (APHRC) internal ethics review committee. We trained the research assistants on research ethics before data collection and held a community entry meeting to inform community members about the study's aims and seek their permission. Participants were informed of the study's objectives, and that their participation was voluntary and anonymous. We obtained written informed consent from participants aged 18 years and older, and for those younger than 18 years, written parental or guardian consent and participant assent were sought. Participants received 300 Kenyan shillings (approximately 2.5 US dollars) as compensation for time loss at the end of the interviews in line with institutional research guidelines and the ethics committee recommendation.

## Positionality statement

Despite the research team's initial concerns about the sensitivity of talking about adolescent pregnancy with the participants, data collection proved interesting, easy, and fruitful. The successful interactions with pregnant and parenting adolescents are attributable to careful considerations made beforehand. We worked with youthful female research assistants who could easily engage with the adolescent girls and understand them. All four research assistants were bilingual and two had experienced adolescent pregnancy. The research assistants were university graduates and could easily relate with the participants due to prior experience in qualitative data collection in the study setting. We avoided male research assistants to create an open and comfortable environment for the all-female adolescent participants, helping to build rapport and keeping the adolescents at ease while sharing sensitive personal narratives.

Both AIA and CWK were principal investigators in the larger study. BNM and CWK were born and raised and have studied and conducted research in Kenya. AIA, on the other hand, has extensive experience in conducting research on adolescent sexual and reproductive health in Kenya and other African countries. These backgrounds offered an advantage in

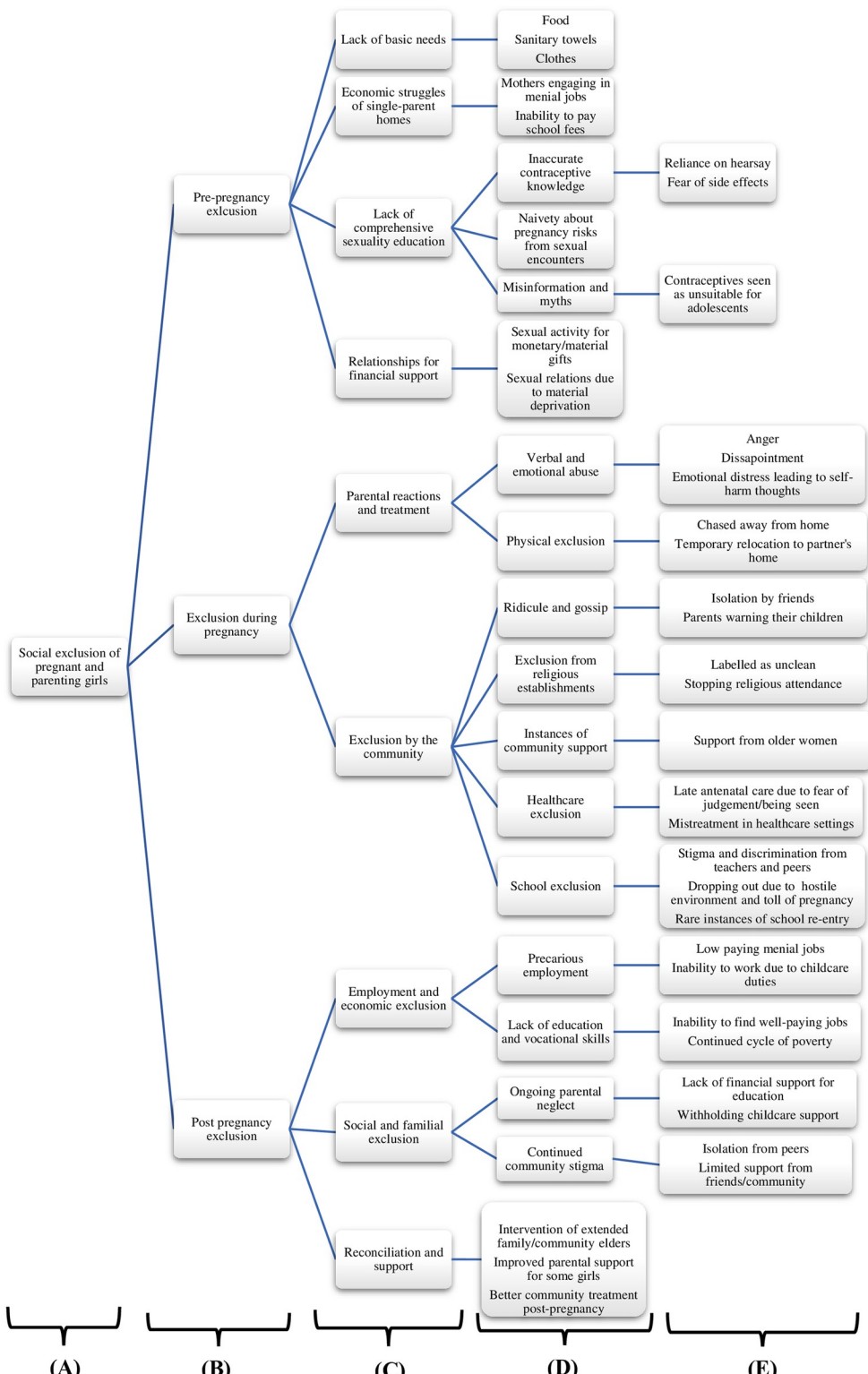

**Fig 2. Coding map.** (A) Study focus (B) The different stages through which social exclusion of girls occurs (C) Core themes of social exclusion identified in the study (D) Sub-themes comprising specific aspects of social exclusion (E) Illustrative examples that provide context for each sub-theme.

comprehending the societal norms, laws, and girls' experiences of social exclusion without skepticism or stereotypes. However, this familiarity posed a risk of reduced objectivity, and potential assumptions based on previous knowledge. The authors mitigated this by valuing the narratives and viewpoints of all participants and avoided imposing their own biases on the collected data.

## Findings

The analysis indicates that adolescents experienced exclusion before, during, and post-pregnancy. We describe these patterns of exclusion in the themes below:

### Poverty and material deprivation pushed adolescent girls to engage in early sex

Almost all the girls interviewed alluded to entering a relationship early with the hope of having their basic needs met. Even those that did not directly express this sentiment somewhat linked their becoming pregnant to poverty and material deprivation. For context, most girls interviewed were from single-parent homes in a low-income urban settlement. They lived with their mothers, who engaged in menial jobs and were unable to adequately provide for their basic needs, including sanitary towels, food, and clothes. Many of them recounted how financial and material deprivation they experienced led them to engage in sexual relations. While some believed men could provide for their needs, others could not resist the advances of men who approached them and offered them monetary and material gifts. A few girls recounted how the lack of school fees truncated their transition from primary to secondary school. They recalled that it was during their stay at home that they entered into dating relationships. A 19-year-old parenting girl stated:

> "I could not go to high school, as my mother had no work and drank alcohol all the time. Getting food was also a problem, that was when I met the father of this child, and we started a relationship until I completed class eight. In the course of the relationship, I got pregnant and that is how we started living together" (IDI participant 20, aged 19, a mother of two, who lives with her husband).

All parents interviewed disclosed that they were facing financial struggles, often relying on credit and low-wage jobs, which made it challenging to afford essentials such as food, school fees, and clothing for their daughters. They explained that economic strain sometimes pushed girls into sexual relationships with men to fulfill these needs. A mother to a girl who got pregnant at 15 years narrated:

> "I do not know whom to blame for this pregnancy because when I am out looking for work, I leave them at home, and I am never sure if they might have plans to meet boys outside. I always bring food home, but there are times when there is none, and I tell them so. Maybe the lack of food at home contributed to them leaving and getting pregnant. Because we eat or sometimes sleep hungry and get used to it. Maybe that is what led them to get pregnant outside there" (IDI parent 1, a mother to a girl who got pregnant aged 15 years)

> "I saw that my coming back from work at night was making my child to be tough headed and so, I stopped going to work at night. I would go in the morning and come back by 4 pm so that I could keep a close watch on her but it was too late because, she was already pregnant the minute she stepped into form two" (IDI parent 2, single mother to a 17-year-old parenting girl).

Other parents explained that their economic hardships and dependence on low paying jobs forced them to live paycheck to paycheck, frequently working long hours, which restricted their ability to effectively supervise their daughters and led them to feel responsible for their pregnancy. A parent to a parenting girl reported:

*"I used to work most of the time. When I left for work in the morning or went for night shifts, she would also leave, and that's when she got pregnant. Initially, I worked morning shifts, returning by 4 pm. Occasionally, I took night shifts, returning the following morning. I noticed my late-night returns were affecting my child's behavior, so I stopped working nights. I switched to morning shifts, returning by 4 pm, hoping to keep a closer eye on her. However, by then, she had already become pregnant"* (IDI parent 2, single mother to a 17-year-old parenting girl).

The typical partners of adolescent girls in the study setting were young men, who worked menial jobs either as boda-boda riders (motorbike), garbage collectors, car washers, casual laborers in construction projects, and plumbers, among others. Because they made some money, they could entice girls into relationships with money and material gifts. The girls interviewed reported lacking food, underwear, rent, body lotion, and washing soap among other basic needs. While some of their parents worked, they earned too little to adequately meet their basic needs. They recounted how going hungry led them to enter relationships with these young men and considered it as their way of finding a solution to their precarious living.

*"So, before I got pregnant, you would find that at home we would not get food to eat because at times my dad would not be working and mum works at the Somalis where she washes clothes for Somalis, and at times, she might come back home with something and other times she might not get anything and so, we would stay hungry for the whole day. Also, we would not get proper undergarments so we would wear undesirable underwear. That's why I saw that instead of living like we were, it would be better that l find some 'way forward' and in the course of that, I found myself in a bigger problem compared to what I was initially experiencing"* (IDI participant 16, aged 17, a mother of one, who lives with her father, mother, and sister).

One girl recalled that it was one episode of her mother being unable to pay their rent resulting in them being kicked out of their one-room shack that made her begin living with her then-boyfriend. It was a few months into living with her boyfriend that she became pregnant. The material and financial deprivation suffered by children of poor parents pushed them into early and unintended pregnancy.

### The exclusion of adolescent girls from accessing sexuality education contributes to their unintended pregnancy

All the girls interviewed linked their lack of accurate knowledge about contraceptive methods to their early and unintended pregnancy. They recounted that they were never taught about contraceptives and their only knowledge—pre-pregnancy—stemmed from what they heard from friends. While they knew contraceptives exist and can help prevent pregnancy, they believed they should not be used by adolescents as they are for older women. Some, having heard rumors that contraceptives could cause health challenges, including weight changes and infertility, did not attempt to use them. With the absence of accurate contraceptive information, girls relied on hearsay, lacking sufficient details, and often mischaracterizing

contraceptive side effects, leading them to shun contraceptive use altogether. A 19-year-old parenting girl narrated:

> "*I knew about contraceptives as my mother would tell me, we have pills, injectables, implants, and coil [intrauterine device or IUD] . . .. However, as you know, there are those rumors where people would tell you they lost weight after using a method among other things. Sometimes they also tell you that those methods are not suitable for people who have never given birth. Therefore, I decided not to use them for fear that I would waste my chance of having a baby in the future*" (IDI, participant 19, aged 19, a mother of one, who lives with her mother).

Most girls only learned about contraceptives during their antenatal care. Their determination to prevent repeat unintended pregnancy pushed them to go for counseling sessions. An 18-year-old parenting girl noted:

> "*I only came to know later, after I got pregnant. That's when people started telling me that I should have used family planning methods like injections. You get injected so that you don't get pregnant, but this was after I had already gotten pregnant*" (IDI, participant 12, aged 18, a mother of one, who lives alone).

In some instances, girls began using contraceptives after their parents intervened following childbirth. A mother narrated taking her daughter for a 3-month family planning injection after childbirth to prevent a repeat pregnancy:

> "*You can have a child in your house who is very polite but when they get out, you may not know what they are doing. Especially these polite girls that are normally quiet, these kids usually take us 'speedy' as they are quiet when they are inside and when they go outside, you may not know where they go to. . .That is why I said no, let me just take her for the injection*" (IDI parent 4, a single mother to a daughter who gave birth at 15 years).

Several girls interviewed recounted that they did not know that having sex for the first time could result in pregnancy. Some even believed that they needed to engage in sex many times before they could become pregnant. Such naivety can be attributed to their lack of accurate sexuality education. The exclusion of girls from accessing accurate information on sexuality and contraception information is by design, as efforts to introduce CSE in schools are resisted.

## Becoming pregnant exacerbates girls' social exclusion

All the girls interviewed reported that their early pregnancy provoked harsh reactions from their parents. For context, all girls were single and living with their parents when they became pregnant, except for one who was cohabiting at the time. When their parents found out about their pregnancy, they were furious, leading them to become verbally and emotionally abusive towards them. In extreme cases, parents angrily chased their pregnant daughters away from home. In one case, one father insisted his daughter leave the house around 10 p.m. and threw her personal belongings out of the house. She moved in with her boyfriend who lived about a kilometer away, but the father came to chase her away from there too, insisting that she could not live anywhere near him. She was later rescued by a teacher who took her in and gave her a job as a domestic help. When asked about how her parents reacted to her pregnancy, she recounted:

*"I started living with my boyfriend, but my father came and asked me to leave, saying he did not want me there. The place was not far from our home, so I was left wondering what my father wanted because he had sent me away from home with my clothes at 10 pm telling me he did not want me and took me to the person that got me pregnant. Now, my father had asked me to leave, yet my maternal side did not want me with the pregnancy saying I did not leave them pregnant and that I should go back to the person who got me pregnant"* (IDI, participant 12, aged 18, a mother of one, who lives alone).

Sometimes the girls themselves chose to move in with their boyfriend because they feared their parents' possible angry reaction to their getting pregnant or in response to the verbal abuse from their parents. For some, moving away from home was temporary and they returned after a while or post-delivery, often when they quarreled with their boyfriends, or due to the ill-treatment from their partners, or due to poorer living conditions with their partners. Some parents did not chase their pregnant daughters away but isolated them, refused to talk to them, or withheld the support they normally rendered, including school fees. This view was captured in the response of an 18-year-old parenting girl:

*"My dad talks to me but not like before. I would walk with him as he held me and even made my hair even while I was already this big but currently, he does not even want to walk with me. Whenever I sit with him in the sitting room he would wake up and leave and he would just pass by my side and ignore my greetings sometimes"* (IDI participant 17, aged 18, a mother of one, who lives with her sister and sister's husband).

One parent recounted her deep distress upon finding out about her daughter's pregnancy. Learning that her daughter had been to a man's house only added to her frustration. She blamed her daughter for getting pregnant and ended up avoiding her. As a single parent who had worked tirelessly to support her daughter's education, she could not understand how she had gotten pregnant despite all her efforts:

*"I was stressed and could not bring myself to speak. I just handed her money for food and left for a friend's place... All I could think about was how I have struggled on my own, using money from washing clothes to send her to school. How did she end up pregnant when I struggled so much for her? ... I do not blame anyone; I just blame her. What was she doing in a man's house?... I do not know what made her go there, yet I had left her some food when I went to work"* (IDI parent 3, 32-year-old mother to a 15-year-old pregnant adolescent girl).

Girls recounted how the ill-treatment from their parents made them become stressed, lose their appetite, and attempt self-harm and suicide. There was a particular girl who sought to buy poison to end her life and stated that it was better to die than continue to suffer emotional abuse from her parents. She later changed her mind with the intervention of her uncle. Parents' anger was born of huge disappointment in their daughters, feelings that they had disgraced the family, and perceived loss of their investment in their daughters' education.

Reconciliation came much later, often after the intervention of extended families and occasionally community elders. Reconciliation led to better parental support, including for their newborn, and improved psychological well-being:

*"I started going outside in December after reconciling with my mother. I told myself that if my mother supported me, then whoever talked ill of me did not matter. So, I started going out,*

*sitting outside by the roadside, and talking to those who wanted to talk to me"* (IDI, participant 20, aged 19, a mother of two, who lives with her husband).

Reconciliation and parental acceptance helped girls decide to keep their pregnancies and boosted their self-confidence. For some girls, parental acceptance assisted them to better deal with stigma from the community and led them to stop isolating themselves.

Girls also faced exclusion from friends and the community. They were ridiculed gossiped about and isolated. Other parents in the community warned their children against associating with them, labeling them as a bad influence. A 19-year-old parenting girl stated how the community's attitudes forced her into isolation:

*"They used to stare as I passed, and I would hear them talk about me as I moved ahead. Some were also laughing at me. . .I was so embarrassed. . . I had to wait until they went back inside before going outside or cover my face so that I would not see them"* (IDI, participant 11, aged 19, a mother of two, who lives with her mother).

Exclusion from the community extended to religious establishments. It was common for girls to report that they stopped attending religious worship after getting pregnant because they were labeled as "unclean" and due to feeling ashamed. A 19-year-old girl who was pregnant at the time of the interview explained:

*"For now, what we believe is that someone is not supposed to go to church when she gets pregnant. First, you need to stop because you are unclean"* (IDI, participant 5, aged 19, currently pregnant, who lives with her mother and siblings).

While exclusion from the community was common, there were a few instances when girls received support in the form of food, a place to stay, and words of encouragement from older women in their communities. This gesture made them persevere in the face of adversity. Often, exclusion from the community was pervasive during pregnancy but got better post-pregnancy, with some girls reporting they received material support, in the form of childcare products and basic commodities. People who initially discriminated against the girls during pregnancy began treating them better and offered advice and support. The excerpt below illustrates this change:

*"Those who used to judge me would now bring themselves close to me and give me advice, like, you did well not to abort the pregnancy and decided to have the baby. It was good that now the baby is big and that I can just leave the baby and go to work. . ."* (IDI, participant 10, aged 17, a mother of one, who lives with her mother and siblings).

Exclusion from their family and community had negative implications for girls' antenatal care attendance. Because of the social exclusion of pregnant girls, it was common for girls to seek antenatal care late and fail to complete the recommended number of visits. Girls feared being seen by people in the community on their way to the hospital and even by health workers. When it was inevitable for them to go out of their homes, they would conceal the pregnancy by wearing oversized clothes. When girls voluntarily or involuntarily moved away from home to stay with their partners, they missed parental guidance on the need to seek antenatal care. A few girls reported that the hospital environment was unfriendly, with health workers questioning and judging them, resulting in them sparingly attending antenatal care.

Most girls faced active and passive exclusion from school. When news about their pregnancy became known to their schoolmates and teachers, they faced stigma, discrimination, gossip, ridicule, and insults. Both parents and the girls reported that girls often refused to go back to school due to fear of facing shame, embarrassment, and stigma from their peers and teachers. Girls also stated that their classmates isolated them, refusing to sit close to them or talk to them.

*"Other students were saying, this girl is pregnant, and some were not talking to me. They said if they sit with me, they will also become pregnant. So, I used to just sit with one of my friends. . ."* (IDI, participant 2, aged 16, a mother of one, who lives with her boyfriend).

The hostile school environment, fear, and shame made those who were in school to stop schooling. In a few cases, some continued with school until the fifth and sixth months but later dropped out of school due to the toll of the pregnancy on them, inability to concentrate, and morning sickness. The interviews with key stakeholders revealed that teachers blame girls for getting pregnant and expel them from school altogether. This view was supported by a representative of a civil society organization:

*"Most of them prefer not to go back to school because of the reception they got from the teachers and students. A teacher tells you, 'You came here to get pregnant. Go take care of the baby"* (KII, civil society organization representative 2, a team leader involved in restoring dignity in the community).

School reentry was rare among adolescent mothers. All but two of the girls interviewed were out of school post-pregnancy. Younger adolescents, aged 15 to 16 years, had either dropped out of primary school or were still in secondary school when they became pregnant. In contrast, among older adolescents aged 17 to 19 years, some had dropped out of primary or secondary school, while others had completed primary education but could not progress to secondary school after getting pregnant. While Kenya has a school reentry policy that unconditionally guarantees school re-entry of adolescent mothers, girls feared returning to school. Even though some girls reported that they would like to return to school, lack of childcare, and financial and parental support hindered them from returning. In some cases, their parents had withdrawn their support for their education:

*"My father did not chase me away but instead, he never wanted to see me. He would not want to hear anything concerning me. He refused to pay my school fees balance. I have been punished since I got pregnant while still in school, I was left to make sure that I pay the school fees balance on my own so that I get the secondary school certificate. That was the punishment given by my father"* (IDI participant 17, aged 18, a mother of one, who lives with her sister and sister's husband).

While some parents would have liked their girls to continue with their education, their jobs made it impossible for them to provide childcare support, as it would mean they stopped working—an untenable option—to take on childcare responsibility to allow their girls to return to school. Without parental support for girls' education post-pregnancy, the likelihood of school reentry was very low. A 17-year-old parenting girl illustrated this in her response:

*"I would like to go back to school but there is no one to take care of my baby. My mum goes to work, my dad also goes to work, and my young sister is in school. My baby is still breastfeeding*

*and I do not have money to pay at some daycare, so I cannot do it currently"* (IDI, participant 16, aged 17, a mother of one, who lives with her father, mother, and sister).

Every parent revealed that their daughters could not continue their education after becoming pregnant. Although they regretted the disruption in their daughters' schooling, they decided against re-enrolling them to punish them for the additional financial burden caused by the pregnancy. They also mentioned the challenges of managing household expenses while supporting their daughters and grandchildren, coupled with the emotional impact of witnessing their children experience early pregnancy, as reasons for not re-enrolling their girls to school. When asked about the prospects of her daughter's education, a mother of a parenting adolescent girl replied:

*"The pregnancy has really affected it. . . by now she would be in form three and about to finish secondary school, and so, this has really derailed her. She can also see how this has derailed her. . .I am also mad because I lost my money because if she is to go back to school, I would have to look for money to pay"* (IDI parent 2, single mother to a 17-year-old parenting girl).

Most girls were out of school, out of training, and out of employment, a situation detrimental to their empowerment and future ability to earn a decent income to care for themselves and their families. While they felt the need to find jobs to better support their babies, their lack of education qualifications and vocational skills meant they were unable to find well-paying jobs:

*"I have been looking for a job and, in most places, they ask for a form four certificate [secondary school completion certificate], in other places, they even ask for your results: like did you get a B plus or a C and then you do not know what to say"* (IDI, participant 13, aged 19, a mother of one, who lives with a female friend).

*"The girls do not have an education; as a result, they cannot get jobs. They can only look for casual jobs at the company on the other side that packages beans, but it is already full. In the end, the community remains in a state of never-ending poverty"* (KII, community leader 3, village elder).

Only a few girls interviewed were employed. However, they worked in precarious jobs—including washing clothes and dishes or as casual laborers in factories—and earned very little. *"The amount of pay from the work is not sufficient. The twisting job, for example, can only give you 100 or 50 shillings per day. Doing laundry can only give you 300 shillings. That money is not enough to buy clothes, food, and other things"* said an IDI participant aged 19. Their childcare duties also limited their ability to work and further worsened their financial struggles. The excerpt below relays these experiences:

*"I can't go to work because of the baby. Even if I got a side hustle, the money cannot pay rent. The person I stay with is not willing to take care of the child sometimes when I get a job to do"* (IDI, participant 7, aged 18, a mother of one, who lives with her uncle).

## Discussion

We examined how passive and active social exclusion hinder adolescent girls from preventing unintended pregnancy. We also explored how adolescent pregnancy further results in girls'

social exclusion and its implications for their health and economic (dis)empowerment. We found that the social exclusion of girls is both passive and active and not confined to the period of pregnancy or after pregnancy but starts even before conception due to a myriad of interconnected factors. Just as Amartya Sen suggests, social exclusion entails a complex interplay of factors that create a cyclical pattern of marginalization and vulnerability for individuals and groups [36]. The findings of this study indicate that the social exclusion of girls is rooted in social stigma, inadequate sexuality education, and socioeconomic challenges.

One of the major findings of this study is that material deprivation and poverty play a significant role in pushing girls into early sexual debut and pregnancy. The girls in this study lived in a low-income urban settlement, mostly with their mothers who often struggled to fulfill their basic needs. As a result, many of the girls entered sexual relationships with older men and boys who could offer them the material things that they lacked in their households. Similar patterns have been observed in Tanzania and South Africa where adolescent pregnancy has been linked with economic vulnerability [63, 64]. An intriguing observation from our research is that the passive social exclusion of parents from earning living wages, coupled with their unstable employment conditions marked by prolonged work hours away from home, leads to an increased vulnerability of their daughters to early and unintended pregnancies. Due to parents working extended hours away from home and earning minimal wages, their daughters frequently remain unattended and unsupervised, which leaves them vulnerable to negative influences. Amartya Sen posits that passive social exclusion happens when groups or individuals face poverty and are unable to access the services and resources needed for their well-being and participation in society [36]. This finding accentuates the need for targeted poverty alleviation interventions and social safety nets that can help address the socioeconomic challenges faced by adolescent girls living in poverty.

We also found that girls become pregnant due to their exclusion from CSE. Comprehensive sexuality education is a name used to distinguish reproductive education that is not limited to abstinence. The adolescents in this study lacked accurate information about contraceptive methods and the consequences of sexual activity. This finding aligns with Ajayi et al. [65] who found that inaccurate information about contraceptives led to nonuse and ultimately, unintended pregnancy among adolescent girls in Western Kenya. This finding also resonates with a previous study indicating that the absence of CSE generates knowledge gaps and misconceptions about sexual and reproductive health among adolescent girls [66]. Adolescents' lack of CSE can be attributed to the activities of the anti-right movement. While the anti-right movement believes that abstinence-based education is the panacea for ending adolescent childbearing, others believe that many young people are already sexually active and abstinence-only education does not meet the needs of such adolescents. However, this result highlights the significance of offering evidence-based and age-appropriate CSE in schools to empower girls with accurate information and encourage responsible sexual conduct.

Consistent with previous studies [34, 67] we found that after becoming pregnant, girls experienced exclusion within their families, communities, and schools. Parents often reacted with emotional abuse, anger, and withdrawal of educational support, forcing some girls to leave their homes. When parents push girls out of their homes, emotionally abuse them, and withdraw their financial and material support, their chances of empowerment are drastically reduced and opportunities to better their lives vanish. Pregnant girls were often gossiped about, avoided, and stigmatized in schools and the community. The girls were also labeled as bad influence, with other parents warning their children against associating with them. In places of worship, they were considered "unclean". Previous studies conducted in Rwanda and Burundi also found that the existence of norms and morals against premarital sex result in stigma, shame, and silence around the sexual acts of young people [17, 68]. The social

exclusion of pregnant girls is perpetuated and justified based on the belief that such girls are promiscuous. It is also considered a punishment for deviating from the norms of saving sex for marriage. This form of social exclusion can initiate feelings of isolation, stress, and even self-harm among girls.

The stigmatization and social exclusion of girls also extended into healthcare settings, with those who were pregnant fearing mistreatment and judgment from healthcare personnel. This fear led them to delay antenatal care visits, potentially risking their health and that of their babies. Comparable findings have been exhibited in different research, highlighting that stigma negatively affects the healthcare-seeking behaviors of pregnant adolescents [20]. This finding infers that addressing healthcare stigma through creating a non-judgmental and supportive environment for pregnant girls is vital to support their timely and full access to antenatal care.

Post-pregnancy, girls encountered significant challenges in returning to school or getting stable employment. Most girls dropped out of school due to financial challenges, childcare responsibilities, lack of childcare support, and fear of judgment by schoolmates and teachers. Our finding on low school return among adolescent mothers post-pregnancy contrasts findings from studies in South Africa, which found that a majority of adolescent mothers returned to school after childbirth [69, 70]. However, in South Africa, parenting girls are eligible for financial support in the form of child grants, facilitating their school return. As such, it is not surprising that most girls in our study failed to return to school post-pregnancy, as this could be attributed to the increased financial burden placed on their parents to support the babies since they were already living in poverty. In line with this finding, the World Health Organization [8] explains that socio-economic vulnerability, fear of negative social experiences in school, and lack of childcare support can limit the educational attainment of adolescent mothers.

Consequently, limited education and lack of vocational training often perpetuate pregnant and parenting adolescents' economic disempowerment through unemployment or work in menial and low-paying jobs. Similar conclusions were drawn from data from the 1970 British Cohort Study, which demonstrates that teenage motherhood has a significant negative long-term effect on hourly wages. At age 42, teenage mothers earned 12% less than other women and 29% less than women who have not had any children [71]. Therefore, comprehensive policies addressing stigmatization are needed to support adolescent mothers in attaining educational and economic empowerment.

## Policy implications

This study highlights the need for targeted poverty alleviation efforts to address the socio-economic challenges of girls living in poverty to reduce their vulnerability to early pregnancy. These efforts can encompass financial assistance for basic necessities like food, school fees, and clothing, combined with access to economic opportunities customized to their needs. For instance, interventions can consider vocational training focused on enhancing their ability to provide services aligned to local economic needs in sectors such as hospitality, or agriculture. Additionally, programs could be designed to facilitate access to microloans, offer business training, and provide mentorship on entrepreneurship among pregnant and parenting adolescent girls.

The findings call for urgent action to eliminate the social stigma around adolescent pregnancy, which can be realized through sensitizing the community, family, healthcare institutions, and schools about prevailing social norms. Community-wide campaigns should be launched to sensitize families, healthcare providers, educators, and community leaders about the harmful effects of social stigma surrounding adolescent pregnancy. In health institutions,

there is need to enhance adolescent-friendly health services that guarantee confidentiality, affordability, and accessibility to a variety of contraceptive methods and services suitable for girls.

This study also emphasizes the significance of implementing CSE in schools and interventions to enhance adolescents' access to contraceptive information for informed decision-making about childbearing. This education should aim to provide accurate knowledge about contraceptive methods, reproductive health, and healthy sexual behavior to empower adolescents to make informed choices.

Lastly, it highlights the need to develop programs to facilitate the reintegration of adolescent mothers into formal education or vocational training. This can involve providing childcare support, flexible schooling options, and financial assistance to mitigate barriers to educational attainment and guarantee their educational and economic empowerment.

## Study strengths and limitations

While this study offers rich insights on social exclusion associated with adolescent pregnancy in urban informal settlements, a careful consideration of its strengths and limitations is essential to interpret and apply the findings effectively.

This study has several strengths. First, employing a qualitative explanatory design facilitated in-depth exploration of participants' lived experiences and perceptions on how social exclusion intersects with early pregnancy and its consequences. The qualitative approach allowed for rich, detailed, and diverse narratives from adolescent girls, parents/legal guardians, and key informants, thus enriching the analysis and findings.

Second, this study offers new insights by highlighting the compounded effect of social exclusion that begins even before conception. Unlike previous research, it uncovers that material deprivation and lack of CSE not only lead to early pregnancy, but also sustain a cycle of marginalization and exclusion during and post-pregnancy. One surprising finding is the extent to which this societal stigma diminishes girls' economic prospects and delays antenatal care. Additionally, unlike some contexts where adolescent mothers receive substantial support to return to school, this study reveals that in low-income settings, the lack of financial and social support is a major factor that significantly hinders their educational and economic recovery post-pregnancy.

Despite its strengths, this study has some limitations. First, our research focuses on pregnant and parenting girls in a low-income urban informal settlement and findings may not be generalizable to pregnant and parenting adolescents in other geographical contexts or rural settings facing different socio-economic conditions. However, the study holds great significance in understanding how girls' social exclusion can lead to adolescent pregnancy and the disempowerment of girls. Future research should investigate the social exclusion experienced by pregnant and parenting girls in rural regions, those from high-income families, and those with disabilities, and compare their experiences with those from low-income urban informal settlements. Second, this study's reliance on purposive sampling might have introduced selection bias and participants who agreed to participate might not have fully represented the diversity of pregnant and parenting girls' social exclusion experiences within the community. Efforts to mitigate this bias included partnering with a community-based organization for recruitment, but inherent limitations in sampling might still exist.

## Conclusion

This study demonstrates that poverty and the contestation of girls' access to CSE lead to unintended pregnancy. It also indicates that adolescent pregnancy further results in girls' social

exclusion with implications for their health and economic (dis)empowerment. Our study shows that poverty, material deprivation, and limited access to CSE are the major contributors to early pregnancies. The social exclusion of adolescent girls extends into their pregnancies across several areas—from their families and communities to places of worship, schools, and healthcare settings, creating a cycle of marginalization. Post-pregnancy, these challenges persist, with girls being excluded from employment and education, contributing to their further social marginalization and disempowerment. To alleviate these challenges, targeted poverty alleviation programs, CSE, and improved access to contraceptive services are essential. Further, supportive healthcare settings, interventions to reduce stigma against pregnant and parenting adolescents, and policies supporting the educational and economic empowerment of adolescent mothers are needed to break this social exclusion cycle and support the well-being of adolescent girls and their children.

## Supporting information

**S1 Text. Interview guides.** These are the interview guides used to conduct interviews with participants. The guides were designed to explore the lived experiences of pregnant and parenting adolescents. Separate guides were developed for different respondent categories, including adolescents, teachers, policymakers, parents/guardians, chiefs, village heads, sub-county officers in charge of children, and healthcare providers. Each guide includes sections with open-ended questions to elicit detailed responses about personal challenges, perceptions, and experiences related to adolescent pregnancy and parenting.
(DOCX)

## Author Contributions

**Conceptualization:** Beryl Nyatuga Machoka, Caroline W. Kabiru, Anthony Idowu Ajayi.

**Data curation:** Beryl Nyatuga Machoka, Anthony Idowu Ajayi.

**Formal analysis:** Beryl Nyatuga Machoka, Anthony Idowu Ajayi.

**Funding acquisition:** Caroline W. Kabiru.

**Investigation:** Beryl Nyatuga Machoka, Caroline W. Kabiru, Anthony Idowu Ajayi.

**Methodology:** Beryl Nyatuga Machoka, Caroline W. Kabiru, Anthony Idowu Ajayi.

**Project administration:** Caroline W. Kabiru, Anthony Idowu Ajayi.

**Supervision:** Caroline W. Kabiru, Anthony Idowu Ajayi.

**Validation:** Beryl Nyatuga Machoka, Caroline W. Kabiru, Anthony Idowu Ajayi.

**Visualization:** Beryl Nyatuga Machoka, Anthony Idowu Ajayi.

**Writing – original draft:** Beryl Nyatuga Machoka.

**Writing – review & editing:** Caroline W. Kabiru, Anthony Idowu Ajayi.

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
