## [Decision Letter · Decision Letter 0]

4 Jun 2024

PGPH-D-24-00309

“My father insisted that I have the baby but not in his house”: Adolescent pregnancy, social exclusion and (dis)empowerment of girls in an urban informal settlement in Kenya.

Dear Dr. Machoka,

Thank you for submitting your manuscript to PLOS Global Public Health. After careful consideration, we feel that it has merit but does not fully meet PLOS Global Public Health’s publication criteria as it currently stands. Therefore, we invite you to submit a revised version of the manuscript that addresses the points raised during the review process.

The manuscript has been evaluated by two reviewers, and their comments are available below.

The reviewers have raised a number of major concerns. They feel the manuscript should outline a clearly-defined research question, and they request improvements to the reporting of methodological aspects of the study. The reviewers also request more depth in the introduction.

Could you please carefully revise the manuscript to address all comments raised? Please also upload your interview guide(s) as "supporting information" files.

We look forward to receiving your revised manuscript.

Kind regards,

Steve Zimmerman, PhD

PLOS Staff Editor

Journal Requirements:

Additional Editor Comments (if provided):

Reviewers' comments:

Reviewer's Responses to Questions

**Comments to the Author**

1. Does this manuscript meet PLOS Global Public Health’s publication criteria? Is the manuscript technically sound, and do the data support the conclusions? The manuscript must describe methodologically and ethically rigorous research with conclusions that are appropriately drawn based on the data presented.

Reviewer #1: Yes

Reviewer #2: Yes

2. Has the statistical analysis been performed appropriately and rigorously?

Reviewer #1: N/A

Reviewer #2: N/A

3. Have the authors made all data underlying the findings in their manuscript fully available (please refer to the Data Availability Statement at the start of the manuscript PDF file)?

Reviewer #1: No

Reviewer #2: Yes

4. Is the manuscript presented in an intelligible fashion and written in standard English?

Reviewer #1: Yes

Reviewer #2: Yes

5. Review Comments to the Author

Reviewer #1: Thank you for your interesting manuscript. This paper explores the important question of the interrelation between early pregnancy and processes of social exclusion. Although I find the approach interesting, I have some major concerns.

Introduction

• In the introduction different aims of the study are formulated (p. 1, p. 5). No clear research question is formulated

• The importance of parents is not sufficiently emphasized

• The argument is very abbreviated to economic factors for early pregnancy. Other factors are certainly relevant here (cf. e.g. results from resilience research), which are not taken into account here.

• The term socioeconomic well-being should be defined.

• You argue, that girls interchange basic needs for sex. What do you exactly mean with basic need.

• In my point of view, it would be helpful to clarify active exclusion for marginalised communities (p. 5). The argumentation in this section is not entirely conclusive.

• Can you please elaborate on the theoretical positioning of both social exclusion framework and intersectionality theory (and in particular the connection between the two theoretical approaches)?

• Please give more information about the situation of adolescent pregnancy in Kenya (especially focusing on mechanisms of social exclusion)

Methods Section

• You describe that the results come from a mixed-method study. What is the connection between qualitative and quantitative data? And why are only qualitative data reported here?

• I would suggest that you use the COREQ-guidelines to report on the qualitative methods. There are some important issues missing (e.g. length of interviews, data saturation, data validation)

• Can you please add the interview guide and the coding tree as a supplemental file?

Results

• The structure of the results section has been chosen appropriately. I would go into more detail on the conditioning factors for social exclusion, precisely because different groups (cf. p. 6) were defined. I suppose, that the experiences of early pregnancy are certainly very different between the subgroups. This would also help to broaden the intersectional view (which was stated in the introduction)

• There are only a few results of the second group of participants. Are there differences and/or similarities between the adolescents and adults?

• The role of parents is only presented from the girls’ perspective. How do the parents see their own role?

• The last quote is not sufficiently integrated into the text

Discussion

• The interrelated factors metioned (p. 13) are not sufficiently emphasised in the results section. In my view, as I said, it makes sense to compare different groups and their experiences in order to illustrate intersections between conditions of social exclusion

• The references to other studies are worked out. But what is really new about this study? What were the surprising findings?

• The discussion point about the anti-right movement remains very vague. What role did that play in the interviews?

• Regarding policy implications: It would be helpful to concretise strategies to improve the girls’ situation.

• Regarding study strengths and limitations: please check the COREQ-guidelines

Minor concerns

• There are minor redundancies in the introductory section.

Reviewer #2: Thank you for the opportunity to review this paper. I must congratulate the authors for the well-written manuscript. Nevertheless, I believe there are still some points for improvement.

Abstract

1. The methods section of the abstract should indicate the study design.

2. Keywords: I would suggest replacing wellbeing with 'Reproductive Health', and delete 'health'.

Introduction and background

3. While the introduction is straightforward, it is quite limiting. The authors have to help readers understand how rife adolescent pregnancy is in Kenya and particularly in slum settlements. Then, they must zoom in to the differences in rural and urban slums so that readers can appreciate why the focus is on urban slum settlement.

4. The authors must dedicate a section to discuss Sen's social exclusion framework. Let us understand what it is, its assumptions and how it fit this study.

Methods

5. Why was qualitative explanatory design used? You should remember that your methodology is also literature to other researchers.

6. How was the study site 'Korogocho' selected? Is that the only urban slum? We need to understand the rationale behind this selection.

7. "Through a community-based organization in the study setting..." Which community-based organization? Does any of the authors work with this organization? If yes, how was reflexivity ensured?

8. How did you select the second and third group of participants?

9. It is not enough to say 'purposively sampled'. Practically, what did you do? This is vital to validate your methodology and inform other researchers.

10. At what point did you stop the interviews? If it was by saturation, then what was the stopping criteria?

11. The authors must specify what they mean by, "Four well-trained research assistants conducted face-to-face interviews". What was their educational level and experience in conducting such research?

12. Was there any training for the research assistants? If yes, how long and what did it cover?

13. At what point what inductive analysis done and what point for deductive analysis?

14. What qualitative research framework informed the thematic analysis?

15. How did you ensure rigor and trustworthiness?

Findings and discussion

16. I entreat the authors to develop a diagram showing how the themes and sub-themes fit into Sen's social exclusion framework.

17. As indicated, providing details of Sen's social exclusion framework would help a lot to better situate the study's findings.

18. "By distilling and synthesizing diverse views of key stakeholders, we provide a detailed analysis of social exclusion associated with adolescent pregnancy". How is this a strength of the study? Is it not what your study intended to achieve?

19. Reflect on the methodology used in this study to beef up the strengths and limitations section.

6. PLOS authors have the option to publish the peer review history of their article (what does this mean?). If published, this will include your full peer review and any attached files.

**Do you want your identity to be public for this peer review?** For information about this choice, including consent withdrawal, please see our Privacy Policy.

Reviewer #1: No

Reviewer #2: No

---

## [Decision Letter · Decision Letter 1]

30 Aug 2024

“My father insisted that I have the baby but not in his house”: Adolescent pregnancy, social exclusion and (dis)empowerment of girls in an urban informal settlement in Kenya.

PGPH-D-24-00309R1

Dear Ms. Machoka,

We are pleased to inform you that your manuscript '“My father insisted that I have the baby but not in his house”: Adolescent pregnancy, social exclusion and (dis)empowerment of girls in an urban informal settlement in Kenya.' has been provisionally accepted for publication in PLOS Global Public Health.

Best regards,

Julia Robinson

Executive Editor

Reviewer Comments (if any, and for reference):

Reviewer's Responses to Questions

**Comments to the Author**

1. If the authors have adequately addressed your comments raised in a previous round of review and you feel that this manuscript is now acceptable for publication, you may indicate that here to bypass the “Comments to the Author” section, enter your conflict of interest statement in the “Confidential to Editor” section, and submit your "Accept" recommendation.

Reviewer #2: All comments have been addressed

2. Does this manuscript meet PLOS Global Public Health’s publication criteria? Is the manuscript technically sound, and do the data support the conclusions? The manuscript must describe methodologically and ethically rigorous research with conclusions that are appropriately drawn based on the data presented.

Reviewer #2: Yes

3. Has the statistical analysis been performed appropriately and rigorously?

Reviewer #2: Yes

4. Have the authors made all data underlying the findings in their manuscript fully available (please refer to the Data Availability Statement at the start of the manuscript PDF file)?

Reviewer #2: Yes

5. Is the manuscript presented in an intelligible fashion and written in standard English?

Reviewer #2: Yes

6. Review Comments to the Author

Reviewer #2: I have reviewed the revised manuscript. The authors have responded satisfactorily to the comments raised. I therefore recommend the acceptance of the paper.

7. PLOS authors have the option to publish the peer review history of their article (what does this mean?). If published, this will include your full peer review and any attached files.

**Do you want your identity to be public for this peer review?** For information about this choice, including consent withdrawal, please see our Privacy Policy.

Reviewer #2: No
